# Antifungal and Antioomycete Activities of a *Curcuma longa* L. Hydroethanolic Extract Rich in Bisabolene Sesquiterpenoids

Adriana Cruz [1,2], Eva Sánchez-Hernández [3,*], Ana Teixeira [1,4], Pablo Martín-Ramos [3], Ana Cunha [1,2] and Rui Oliveira [1,2]

[1] Department of Biology, School of Sciences, University of Minho, Campus de Gualtar, 4710-057 Braga, Portugal; cruzadriana73@gmail.com (A.C.); anaspereirateixeira@gmail.com (A.T.); accunha@bio.uminho.pt (A.C.); ruipso@bio.uminho.pt (R.O.)

[2] Centre of Molecular and Environmental Biology (CBMA), University of Minho, Campus de Gualtar, 4710-057 Braga, Portugal

[3] Department of Agricultural and Forestry Engineering, ETSIIAA, Universidad de Valladolid, Avenida de Madrid 44, 34004 Palencia, Spain; pmr@uva.es

[4] Centre for Research and Technology of Agro-Environmental and Biological Sciences (CITAB), Inov4Agro, University of Trás-os-Montes and Alto Douro (UTAD), Quinta de Prados, 5000-801 Vila Real, Portugal

[*] Correspondence: eva.sanchez.hernandez@uva.es

**Abstract:** *Curcuma longa*, known for its anti-inflammatory, antioxidant, and antimicrobial properties, has been a staple in traditional medicine for centuries. In the pursuit of natural alternatives to synthetic preservatives, the extracts of *C. longa* have garnered attention for their efficacy in extending shelf life and preventing the spoilage of diverse agricultural products. This study aims to assess the antifungal and antioomycete activities and plant protection capabilities of a hydroethanolic *C. longa* extract as a natural product against crop pathogens. The phytochemical profile of the *C. longa* extract was elucidated through gas chromatography-mass spectrometry. The extract exhibited a richness in bisabolene sesquiterpenoids, notably (+)-β-turmerone, α-turmerone, (+)-(S)-*ar*-turmerone, and included minor phytoconstituents, such as α-atlantone, γ-curcumene, zingiberene, isoelemicin, and gibberellin A3. Radial growth inhibition assays demonstrated the *C. longa* extract's effectiveness against various phytopathogenic fungi, including *Botrytis cinerea*, *Colletotrichum acutatum*, and *Diplodia corticola*, as well as the oomycetes *Phytophthora cactorum* and *Phytophthora cinnamomi*. In ex situ tests, the *C. longa* extract demonstrated remarkable protection capabilities for *Malus domestica* excised stems against *P. cinnamomi*-induced necrosis. Furthermore, the *C. longa* extract exhibited non-toxicity towards lettuce seed germination and did not impact early lettuce seedling growth, indicating its safety for crop protection. These findings contribute to expanding the potential applications of *C. longa* as an antimicrobial agent, particularly for safeguarding economically significant trees against the destructive oomycete *P. cinnamomi*.

**Keywords:** apple tree; *Curcuma longa* extract; biorational; antifungal activity; antioomycete activity; excised stem protection

## 1. Introduction

In contemporary agriculture, a primary challenge confronting the sector is the effective control of plant diseases, leading to annual losses of approximately 10–16% of crops [1]. The impact of plant diseases has been exacerbated by modern farming practices, such as the utilization of genetically identical crop plants, extensive reliance on plant cultivars susceptible to specific pathogens, excessive use of mineral fertilizers, the emergence and proliferation of microbial resistance, and the absence of suitable crop rotation [2].

In this context, pesticide usage is integral to safeguarding agricultural crops from pests and diseases [3]. Nevertheless, the continuous exposure of crops to mixtures of bioactive synthetic compounds raises significant environmental concerns, resulting in the

contamination of terrestrial and aquatic ecosystems and posing risks to environmental and human health. Furthermore, the rapid evolution of pathogens presents ongoing challenges, leading to the emergence of resistances and rendering plants susceptible to diseases [4]. Consequently, epidemics persist, causing substantial yield losses.

Fungal and oomycete pathogens, in particular, stand out as the two major groups of pathogenic organisms responsible for the most destructive plant diseases, affecting both natural and production systems [5,6]. These phytopathogens include, for instance, *Botrytis cinerea* Pers., causing gray mold disease [7]; *Colletotrichum acutatum* Simmonds, causing anthracnose disease [8]; *Diplodia corticola* A.J.L. Phillips, A. Alves and J. Luque, causing Botryosphaeria canker [9]; *Fusarium culmorum* (W.G.Smith) Saccardo, causing *Fusarium* stem and root rot (FRR) and *Fusarium* head blight (FHB; [10]); *Phytophthora cactorum* (Lebert and Cohn) J. Schröt; and *Phytophthora cinnamomi* Rands, causing root rot diseases [11]

Acknowledging the severity of these challenges, researchers, policymakers, and agricultural communities worldwide are actively collaborating to explore eco-friendly alternatives, advocate for sustainable farming practices, and promote awareness regarding the responsible use of agrochemicals [12], a key alliance for success. Encouraging alternatives include the utilization of biological control agents and their secondary metabolites, along with plant extracts and phytochemicals exhibiting antifungal activity [4,13].

Natural products represent valuable resources of bioactive compounds with significant bioactivities and a high potential for the discovery of novel drugs [14]. Approximately 60% of small-molecule drugs introduced from 1981 to 2019 originated from natural products [14]. Plants, in particular, are renowned for their biological properties and unparalleled chemical diversity, allowing them to synthesize metabolites, such as phenolic compounds, flavonoids, coumarins, saponins, terpenes, and alkaloids, with protective functions against abiotic and biotic stresses [15–18].

*Curcuma longa* L., commonly known as turmeric, stands out as a plant species abundant in natural compounds, showcasing diverse bioactivities. It holds great promise not only in the food industry, but also as a potential pharmacological agent in the cosmetic, pharmaceutical, and agriculture sectors [19,20]. The primary phytochemical constituents of turmeric extracts are diarylheptanoids, including curcuminoids [21], along with essential oils [22–24]. Acting as free radical scavengers with reducing power, these compounds serve as antimicrobials against a broad spectrum of food-borne and food-spoilage bacteria and fungi [25]. They have also garnered attention for crop protection as natural substitutes for synthetic fungicides, insecticides, and herbicides [25,26]. For instance, curcuminoids have been successfully employed to prevent fruit and vegetable rot in strawberries, bananas, avocados, and papayas [27–29].

Hence, the primary objective of this study is to investigate a natural antifungal agent capable of effectively replacing conventional fungicides while mitigating the associated drawbacks, especially against the aforementioned filamentous fungi and oomycetes that pose significant threats to global crop yields. Our aim is to discern the principal constituents of a hydroethanolic extract of *C. longa* (CE) through Fourier-transform infrared (FTIR) spectroscopy and gas chromatography–mass spectrometry (GC–MS) analyses, and to assess its antimicrobial efficacy against those fungi and oomycetes, laying the foundation for the development of a crop-protecting product targeted at these phytopathogens.

## 2. Materials and Methods

### 2.1. Microorganisms, Culture Media, and Growth Conditions

In this investigation, the phytopathogens encompassed the filamentous fungi *B. cinerea* (provided courtesy of Richard Breia and Hernâni Gerós from the Department of Biology, University of Minho, Braga, Portugal), *C. acutatum* (provided by Pedro Talhinhas from the School of Agriculture, University of Lisbon, Lisbon, Portugal), *D. corticola* (supplied by Ana Cristina Esteves from the Centre for Environmental and Marine Studies, CESAM, University of Aveiro, Aveiro, Portugal), and *F. culmorum* (CECT 20493, obtained from the Spanish Type Culture Collection, Valencia, Spain). Additionally, the filamentous oomycetes

included *P. cactorum* (CRD Prosp/59, provided by the Aldearrubia Regional Diagnostic Center–Junta de Castilla y León, Salamanca, Spain) and *P. cinnamomi* (provided by the Calabazanos Forest Health Center–Junta de Castilla y León, Palencia, Spain). Isolates of filamentous fungi and oomycetes were cultured on potato dextrose agar (PDA; BioLife Italiana S.r.l., Milan, Italy) medium. Stock cultures were established by placing an 8 mm disc of mycelium at the center of PDA plates, followed by incubation at 25 °C in the dark. Once the organisms reached the edge of the Petri dish, they were stored at 4 °C.

### 2.2. Plant Material and Extraction Procedure

Turmeric, derived from the ground rhizome, was initially procured in a powdered form from a commercial supplier. Five grams of the plant material were subjected to extraction with 30 mL of an 80% ($v/v$) ethanolic solution in a water bath at 60 °C for 30 min, protected from sunlight. The resultant extract underwent filtration and centrifugation (5000 rpm, 10 min; Eppendorf 5804R, Eppendorf, Hamburg, Germany). The solvent in the supernatant was evaporated using a rotavapor (60 °C, 100 rpm; BÜCHI Rotavapor® R-100, Büchi Labortechnik AG, Flawi, Switzerland), lyophilized, and stock solutions of *C. longa* extract (CE) at 200 mg·mL$^{-1}$ were prepared utilizing 80% ($v/v$) ethanol. These solutions were stored at −20 °C, in the dark, until further use.

### 2.3. Extract Characterization

Prior to the characterization, the CE underwent filtration via Whatman No. 1 paper and was then freeze-dried, resulting in a solid residue. The infrared vibrational spectrum of the freeze-dried extract was measured using a Nicolet iS50 Fourier-transform infrared (FTIR) spectrometer (Thermo Scientific; Waltham, MA, USA) with an attenuated total reflectance (ATR) system. The measurement was in the range of 400–4000 cm$^{-1}$, with a resolution of 1 cm$^{-1}$. The resulting spectrum was obtained by combining 64 scans. For the gas chromatography–mass spectrometry (GC–MS) analysis, performed at the Research Support Services of Universidad de Alicante, the freeze-dried extract was dissolved in HPLC-grade methanol (chosen as a solvent due to its volatility and suitability for a wide range of compound analyses) to yield a 5 mg·mL$^{-1}$ solution. This solution was then filtered again. The analysis was conducted using an Agilent Technologies (Santa Clara, CA, USA) 7890A gas chromatograph coupled to a 5975C quadrupole mass spectrometer. The chromatography conditions were as follows: injection volume = 1 μL; injector temperature = 280 °C (in splitless mode); initial oven temperature = 60 °C for 2 min, followed by a ramp of 10 °C·min$^{-1}$ to a final temperature of 300 °C for 15 min. An Agilent Technologies HP-5MS UI column with a length of 30 m, diameter of 0.250 mm, and film thickness of 0.25 μm was used for compound separation. The mass spectrometer conditions were as follows: electron impact source temperature = 230 °C; quadrupole temperature = 150 °C; and ionization energy = 70 eV. The identification of components was based on the comparison of their mass spectra and retention times with the National Institute of Standards and Technology (NIST11) database.

### 2.4. Phytotoxicity Assay with a Lettuce Model

Lettuce (*Lactuca sativa* L.) seeds of the 'Maravilha das 4 Estações' variety (Casa César Santos, lot C22373) were employed as the plant model for phytotoxicity assays. The seeds were disinfected by immersing them in 5% commercial bleach for 20 min, with intermittent manual agitation. Under aseptic conditions, the seeds were rinsed three times with sterile deionized water. After drying, they were evenly distributed across Petri dishes (20 seeds per plate), each containing solid basal Murashige and Skoog (MS) medium supplemented with 2% sucrose and 250 or 500 μg·mL$^{-1}$ of CE or 0.2% ($v/v$) ethanol (solvent control). The seeded plates were maintained at 24 °C, with a 16 h photoperiod and a light intensity of 50–70 μmol m$^{-2}$·s$^{-1}$. After 7 days, the germination percentage, number of expanded cotyledons, presence of the epicotyl apex ($\geq$2 mm), number of leaves, and root length of the seedlings were evaluated.

### 2.5. In vitro Antifungal and Antioomycete Activities

The antimicrobial potential of CE was evaluated by mycelium growth inhibition tests of phytopathogenic fungi and oomycetes. Briefly, 250, 500, or 1000 µg·mL$^{-1}$ of CE or 80% (*v/v*) ethanol (solvent control, to ascertain that any observed antifungal activity resulted from the extract itself and not the solvent used; the same volume as the CE treatment at the highest concentration) were incorporated into a PDA medium, and an 8 mm diameter disc of mycelium of the phytopathogenic agent was placed at the center of the Petri dishes. The plates were incubated at 25 °C in the dark and the mycelial growth diameter was measured every three days until the mycelium of each microorganism in the negative control reached the edges of the Petri dish. The percentage of inhibition was calculated as the percent growth inhibition compared to the negative control (0% of inhibition), according to Equation (1):

$$\text{Inhibition}(\%) = \frac{dc - de}{dc} \times 100, \tag{1}$$

where *dc* represents the mean mycelial growth diameter of the negative control and *de* denotes the mean mycelial growth diameter of each replicate in the CE treatment.

### 2.6. Protection Tests on Artificially Inoculated Excised Stems

In compliance with the restrictions governing in vivo assays involving *P. cinnamomi*, the investigation explored into the effectiveness of the *C. longa* extract at its calculated minimum inhibitory concentration (MIC) of 3000 µg·mL$^{-1}$ through the artificial inoculation of excised stems under controlled laboratory conditions. The inoculation adhered to the procedure outlined by Matheron and Mircetich [30], with modifications as detailed in [31].

In brief, the excised stems of healthy *Malus domestica* var. 'Golden', 'Reinette', and 'Starking Delicious', each with a diameter of 1.5 cm, were sectioned into 10 cm-long pieces using a sterilized grafting knife. Immediately after the excisions, the stem segments were wrapped in moistened sterile absorbent paper. In the laboratory, the freshly excised stem segments underwent sequential immersions: initially in a 3% NaClO solution for 10 min, followed by 70% ethanol for 10 min, and then rinsed thoroughly four times with distilled water. This process aimed to eliminate superficial contaminants in the tissue.

Thirty stem segments per variety were soaked for 1 h in distilled water to serve as controls (15 for the positive control and 15 for the negative control). Concurrently, the remaining stem segments (*n* = 15 segments/variety) were soaked for 1 h in CE at the calculated MIC concentration. To enhance the moistening and penetration of the treatment into the bark, a coadjuvant (Alkir®, 1% *v/v*; De Sangosse Ibérica, Quart de Poblet, Valencia, Spain) was added to the extract (and to the distilled water, in the case of the controls). Following soaking, the stem pieces were allowed to dry, and the bark was meticulously removed with a scalpel to reveal the cambium.

The exposed bark was then placed on an agar Petri dish and, in the cases of the positive control and treated samples, it was inoculated by placing a plug (diameter = 5 mm) from the margin of a one-week-old PDA culture of *P. cinnamomi* at the center of the inner surface of the bark. Post-inoculation, the stem segments were incubated in a humid chamber for 4 days at 24 °C and 95–98% relative humidity. The efficacy of the treatments was assessed by measuring the lengths of the cankers that developed at the inoculation sites. Finally, the oomycete was re-isolated from the lesions and morphologically identified to fulfill Koch's postulates.

### 2.7. Statistical Analysis

The statistical analysis and graphical representation were conducted using GraphPad Prism version 8.4.2 for Windows (GraphPad Software, San Diego, CA, USA). The results are presented as the mean ± standard deviation (SD) of at least three independent replicas, except for the *in vitro* phytotoxicity assay, for which four replicates were performed per condition. The statistical assumptions of normality and homoscedasticity were evaluated using the Shapiro–Wilk test and the Levene and Bartlett tests, respectively. Based on these

assessments, one-way ANOVA and Dunnett's tests for multiple comparisons between the treatment groups and the control were applied to the *in vitro* and phytotoxicity data. The level of significance (*p*-value) for each test is indicated in the figures as follows: ns ($p > 0.05$) denotes non-significant, * ($0.01 < p \leq 0.05$) denotes significant, ** ($0.001 < p \leq 0.01$) denotes very significant, *** ($0.0001 < p \leq 0.001$) denotes highly significant, and **** ($p \leq 0.0001$) denotes extremely significant. For the comparisons of necrosis lengths in the ex situ assays on excised stems, the Kruskal–Wallis non-parametric test was employed, followed by the Conover–Iman post hoc test for multiple pairwise comparisons.

## 3. Results

### *3.1. CE Chemical Characterization*

#### 3.1.1. ATR–FTIR Vibrational Characterization

The ATR–FTIR spectrum of the hydroethanolic turmeric extract (Figure S1, Table S1) revealed stretching vibrations in the range of 3200–3500 cm$^{-1}$ originating from O–H groups. Stretching vibrations at 1623 cm$^{-1}$ were mainly due to the overlapping stretching vibrations of alkenes (C=C) and carbonyl (C=O). A very intense band at 1509 cm$^{-1}$ indicated mixed vibrations, including stretching carbonyl bond vibrations ν(C=O), in-plane bending vibrations around aliphatic δ CC–C, δ CC=O, in-plane bending vibrations around the aromatic δ CC-H of keto and enol configurations, and stretching vibrations around the aromatic νCC bonds of keto and enolic forms. Additionally, there was a C=C aromatic stretching vibration at 1429 cm$^{-1}$. Finally, an intense band at 1267 cm$^{-1}$ was attributed to the bending vibration of the phenolic band ν(C–O) [32]. The identified functional groups were consistent with the presence of the chemical constituents identified in the aqueous ammonia extract by GC−MS, discussed below.

#### 3.1.2. GC–MS Analysis

The main phytochemicals identified in the CE chromatogram (Figure 1) were (+)-*β*-turmerone (28.9%), *α*-turmerone (13.3%), (+)-(S)-*ar*-turmerone (15.3%), and *α*-atlantone (3.0%), represented in Figure S2. Minor phytoconstituents included *γ*-curcumene (0.34%), zingiberene or *α*-sesquiphellandrene (0.43%), isoelemicin (1.48%), and gibbellerin A3 (0.55%). A comprehensive list of all the constituents identified by GC–MS is presented in Table 1. Detailed mass data are presented in Table S2.

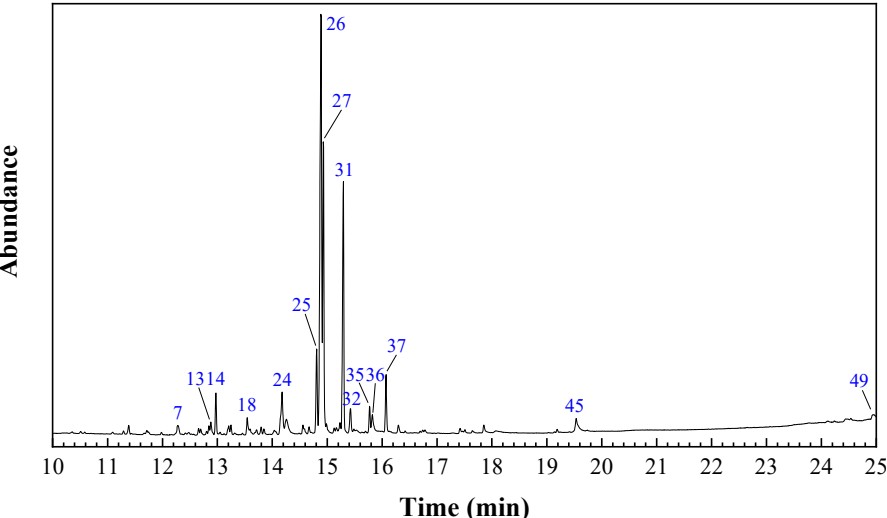

**Figure 1.** GC–MS chromatogram of *Curcuma longa* hydroethanolic extract. The main peaks (as per Table 1) are labeled in blue.

**Table 1.** Chemical species identified in *C. longa* hydroethanolic extract.

| Peak | RT (min) | Area (%) | Chemical Species | Qual |
|------|----------|----------|------------------|------|
| 1 | 10.5118 | 0.1164 | Ethanone, 1-(3-methoxyphenyl)- | 45 |
| 2 | 11.0935 | 0.1303 | Phenol, 2-methoxy-4-(2-propenyl)- (or eugenol) | 90 |
| 3 | 11.3902 | 0.4714 | 3,5,7-trimethyl-2*E*,4*E*,6*E*,8*E*-decatetraene | 43 |
| 4 | 11.7167 | 0.2603 | Isobenzofuran | 43 |
| 5 | 11.7463 | 0.1743 | Benzoic acid, 3,5-dimethyl-, (3,5-dimethylphenyl)methyl ester | 55 |
| 6 | 11.9837 | 0.1657 | 2-Methylbicyclo[4.3.0]nona-2,9-diene dimer | 35 |
| 7 | 12.2805 | 1.0108 | Phenol, 2-methoxy-4-(1-propenyl)- | 83 |
| 8 | 12.4170 | 0.1431 | Oxime-, methoxy-phenyl- | 38 |
| 9 | 12.4823 | 0.2227 | Hydrazinecarboxamide, 2-(1-phenylethylidene)- | 50 |
| 10 | 12.6604 | 0.3421 | *γ*-curcumene | 93 |
| 11 | 12.7019 | 0.3560 | Benzene, 1-(1,5-dimethyl-4-hexenyl)-4-methyl- | 99 |
| 12 | 12.8503 | 0.4319 | Zingiberene | 90 |
| 13 | 12.8859 | 0.8048 | 3,5-dimethoxy-2-methylnaphthalene | 86 |
| 14 | 12.9749 | 1.9900 | 1-phenyl-2-(*p*-tolyl)-propane | 72 |
| 15 | 13.0521 | 0.1909 | 1*H*-Benzocycloheptene, 2,4a,5,6,7,8-hexahydro-3,5,5,9-tetramethyl-, (*R*)- | 38 |
| 16 | 13.2124 | 0.6186 | *β*-Sesquiphellandrene | 98 |
| 17 | 13.2479 | 0.4825 | Dispiro[2.6.2.5]undecane, 10-methylen- | 50 |
| 18 | 13.5447 | 1.4773 | *Trans*-isoelemicin | 90 |
| 19 | 13.5981 | 0.4081 | 3-Acetyl-2-methyl-4-phenylfuran | 50 |
| 20 | 13.7169 | 0.4141 | Acetic acid 2-acetylamino-phenyl ester | 58 |
| 21 | 13.7999 | 0.4219 | (*E*)-1-ethylidene-4,5,8-trimethyl-1,2,3,4-tetrahydronaphthalene | 90 |
| 22 | 13.8474 | 0.3407 | 3,4-Dimethoxyphenylacetone | 68 |
| 23 | 14.0374 | 0.4407 | 3,4-Dimethylbenzyl isothiocyanate | 42 |
| 24 | 14.1798 | 3.4372 | Benzene, 1-ethyl-4-(2-methylpropyl)- | 74 |
| 25 | 14.8089 | 4.4816 | Benzene, 1,2,4-trimethyl- | 43 |
| 26 | 14.8920 | 28.9413 | *β*-Tumerone | 96 |
| 27 | 14.9336 | 15.3193 | *ar*-Tumerone | 90 |
| 28 | 15.1295 | 0.3430 | 2,3,3a,8a-Tetrahydro-2,4-dihydroxy-6-methoxy-furo[2,3-b]benzofuran | 83 |
| 29 | 15.1710 | 0.4089 | 3(5)-(4′-Methylphenyl)- 4-amino-5(3)-ethylaminopyrazole | 83 |
| 30 | 15.2363 | 0.7148 | Pyridine, 4-[(3-methoxyphenyl)methyl]- | 58 |
| 31 | 15.2956 | 13.2673 | *α*-Tumerone | 91 |
| 32 | 15.4262 | 1.6288 | Tetradecanoic acid, methyl ester | 96 |
| 33 | 15.4856 | 0.3040 | 1,3-Cyclohexanedione, 2,2,5,5-tetramethyl- | 43 |
| 34 | 15.5330 | 0.4373 | 3,7,7-Trimethyl-1-(3-oxo-but-1-enyl)-2-oxa-bicyclo[3.2.0]hept-3-en-6-one | 38 |
| 35 | 15.7764 | 1.4119 | 1-Formyl-2-methoxybenzene | 72 |
| 36 | 15.8239 | 1.8154 | Tetradecanoic acid | 99 |
| 37 | 16.0732 | 2.9617 | (+)-*α*-Atlantone | 68 |
| 38 | 16.2987 | 0.6141 | 1-Isopropenyl-3,3-dimethyl-5-(3-methyl-1-oxo-2-butenyl)cyclopentane | 27 |
| 39 | 16.4233 | 0.1388 | 2-Methylthio-3,4-dihydronaphtho[2,1-c]thiophene | 53 |
| 40 | 16.6964 | 0.0922 | 2*H*-1-Benzopyran, 3,5,6,8a-tetrahydro-2,5,5,8a-tetramethyl-, *cis*- | 44 |

**Table 1.** *Cont.*

| Peak | RT (min) | Area (%) | Chemical Species | Qual |
|------|----------|----------|------------------|------|
| 41 | 16.7438 | 0.1690 | 7(1*H*)-Quinolinone, octahydro-4a-(2-propenyl)-, *trans*-(.+-.)- | 35 |
| 42 | 17.4680 | 0.1231 | Methyl 3,5-bis(ethylamino)benzoate | 30 |
| 43 | 17.5095 | 0.2790 | Hexadecanoic acid, methyl ester | 96 |
| 44 | 17.8538 | 0.6914 | *n*-Hexadecanoic acid | 99 |
| 45 | 19.5394 | 1.7381 | 9-Octadecenoic acid (*Z*)- | 99 |
| 46 | 19.7412 | 0.3431 | 1*H*-Indole-3-carboxylic acid, 5-hydroxy- | 44 |
| 47 | 24.1214 | 0.5540 | Gibberellin A3 | 43 |
| 48 | 24.2401 | 0.1306 | 6 Methyl-2 phenylindole | 38 |
| 49 | 24.9524 | 1.0139 | Hexahydropyridine, 1-methyl-4-[4,5-dihydroxyphenyl]- | 46 |

RT = retention time; Qual = quality of resemblance.

### 3.2. In Vitro Antimicrobial Activity

As illustrated in Figure 2, the CE significantly reduced the mycelium growth levels of *B. cinerea* (Figure 2a) and *P. cinnamomi* (Figure 2f) at all tested concentrations. For *C. acutatum* (Figure 2b), *D. corticola* (Figure 2c), and *P. cactorum* (Figure 2e), significance was attained with concentrations of 500 and 1000 μg·mL$^{-1}$ CE. Except for *F. culmorum*, where no significant differences were observed by the end of the experiment (Figure 2d), a noticeable trend of mycelium diameter decrease was observed with increasing CE concentrations compared to the control, indicating a dose–response effect (Figure 2g).

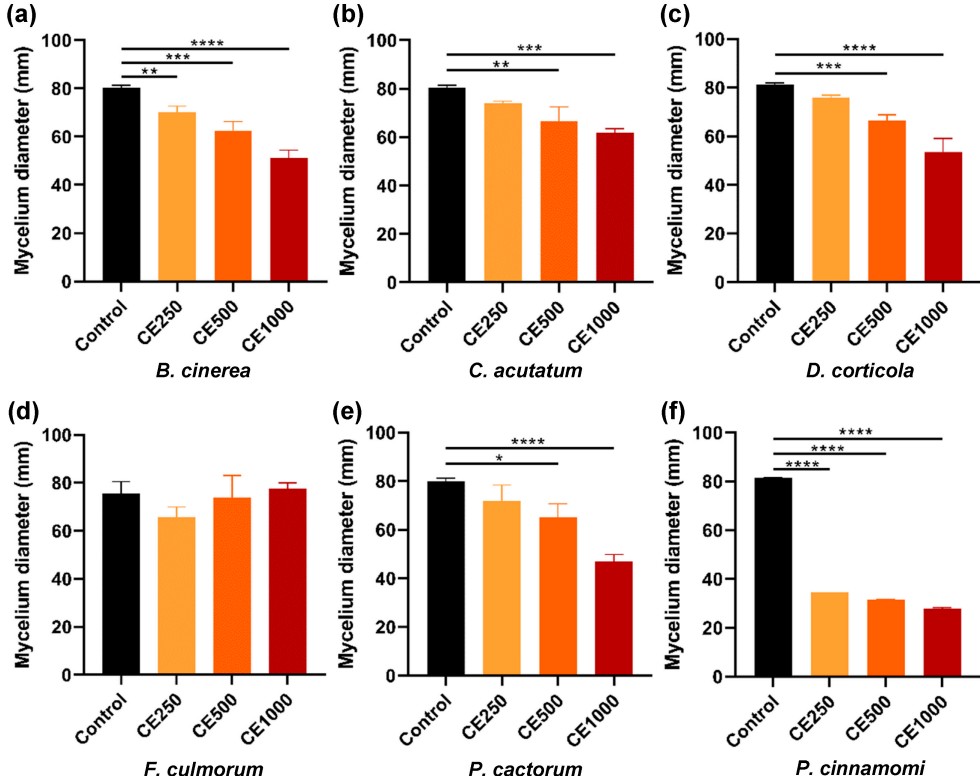

**Figure 2.** *Cont.*

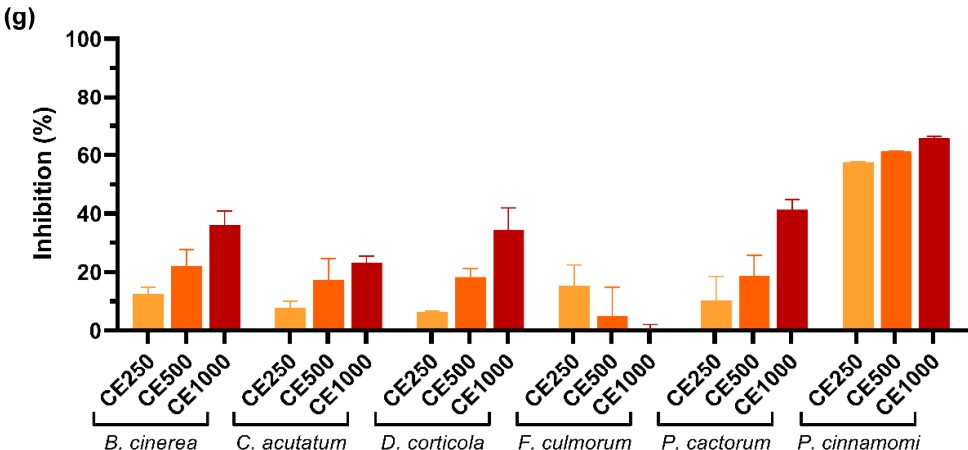

**Figure 2.** *Curcuma longa* hydroethanolic extract (CE) antimicrobial activity against *Botrytis cinerea*, *Colletotrichum acutatum*, *Diplodia corticola*, *Fusarium culmorum*, *Phytophthora cactorum*, and *Phytophthora cinnamomi*. The mycelium diameter (mm; **a–f**) represents the diameter measured on the final day of the assay for each tested microorganism: (**a**) *B. cinerea* (6 days), (**b**) *C. acutatum* (20 days), (**c**) *D. corticola* (10 days), (**d**) *F. culmorum* (17 days), (**e**) *P. cactorum* (24 days), and (**f**) *P. cinnamomi* (6 days) when exposed to 250 (CE250), 500 (CE500), or 1000 (CE1000) µg·mL$^{-1}$ CE or 80% (*v/v*) ethanol (control; the same volume as the highest CE concentration). The mycelium growth inhibition (%; **g**) illustrates the inhibition reached on the last day of the assay for all tested fungi and oomycetes when exposed to the same concentrations. Results are expressed as the mean ± SD of three independent experiments; one-way ANOVA, followed by the Dunnett's test for multiple comparisons. The level of significance is indicated as follows: * ($0.01 < p \leq 0.05$) denotes significant, ** ($0.001 < p \leq 0.01$) denotes very significant, *** ($0.0001 < p \leq 0.001$) denotes highly significant, and **** ($p \leq 0.0001$) denotes extremely significant.

Remarkably high growth inhibitions were achieved against *P. cactorum*, exceeding 40% with 1000 µg·mL$^{-1}$ CE (Figure 2e), and against *P. cinnamomi* at all tested concentrations, reaching approximately a 65% inhibition with the highest concentration of CE (Figure 2f). Notably, with *P. cinnamomi*, the most sensitive microorganism tested against CE, the growth inhibition remained constant throughout the experiment (6 days), a behavior similar to that observed for *B. cinerea* (6 days) and *P. cactorum* (24 days).

### 3.3. In Vitro Germination and Early Growth of Lettuce Seedlings

To ensure the specificity of the target and efficient protection of crops, it was imperative to assess the absence of phytotoxicity. In this regard, an *in vitro* bioassay was conducted using *Lactuca sativa* L. as a plant model. Lettuce seeds were exposed to concentrations of 250 or 500 µg·mL$^{-1}$ of CE for a duration of 7 days, and various parameters of the germination process were monitored.

Except for the root length, the CE did not have a significant impact on the germination rate, cotyledon expansion, presence of the apex, or the number of leaves in the seedlings at both tested concentrations (Figure 3a,c–e). Regarding the root length, a significant increase was observed in comparison to the solvent control at both tested concentrations, especially at the lowest one (Figure 3b). These findings suggest that CE has no relevant inhibitory effects on the early growth of lettuce seedlings, except for the development of roots, where a promotion of growth is suggested.

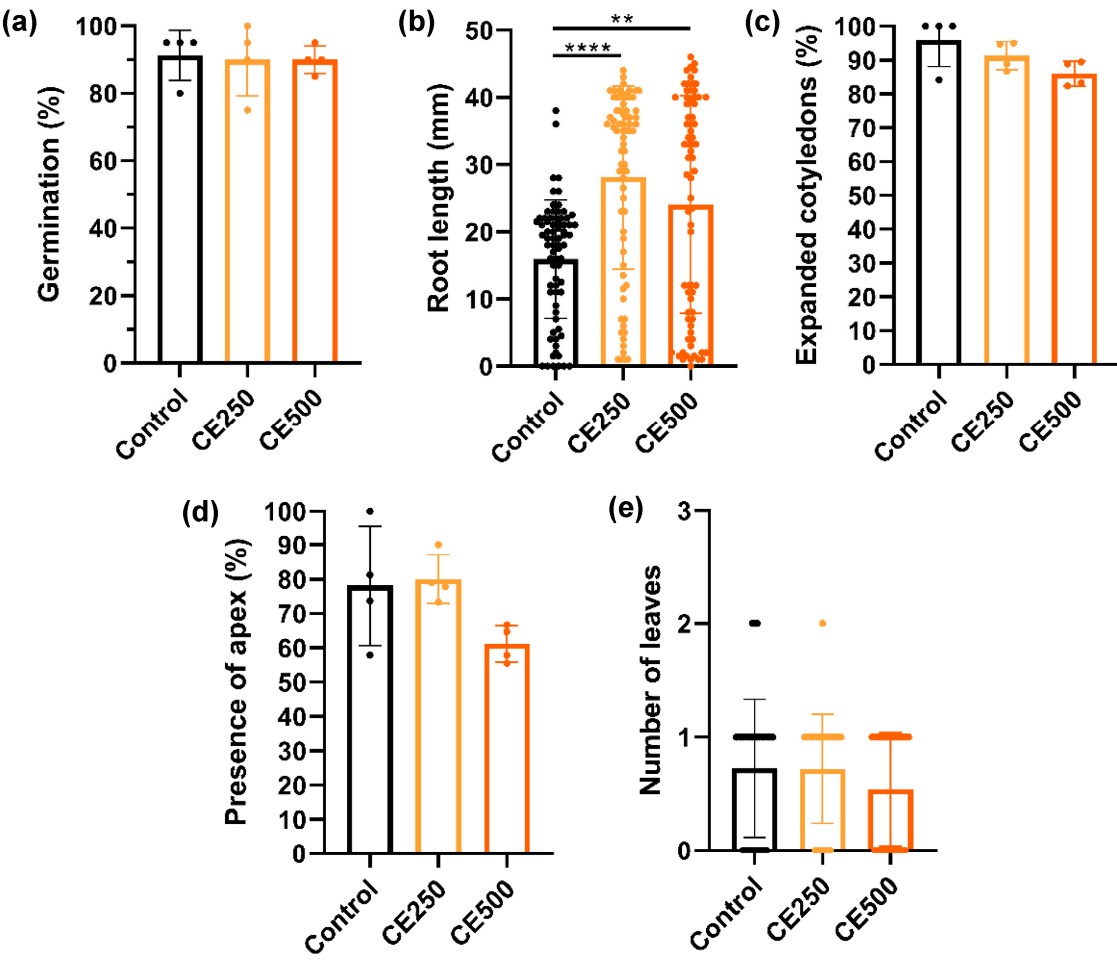

**Figure 3.** *Curcuma longa* hydroethanolic extract (CE) toxicity assessment of *Lactuca sativa* L. seeds and early growth seedlings. Seeds were exposed to 250 µg·mL$^{-1}$ CE (CE250), 500 µg·mL$^{-1}$ CE (CE500), or 80% (*v/v*) ethanol (control) for seven days, and the effect on (**a**) germination, (**b**) root length, (**c**) expanded cotyledons, (**d**) presence of apex, and (**e**) number of leaves were assessed. Data are presented as the means of four independent replicates ± standard deviations (SDs). One-way ANOVA was conducted, followed by the Dunnett's test for multiple comparisons. The level of significance (*p*-value) is indicated as follows: ** (0.001 < *p* ≤ 0.01) denotes very significant and **** (*p* ≤ 0.0001) denotes extremely significant.

### 3.4. Protection of Excised Stems against P. cinnamomi

The results depicted in Figure S3 and Table 2 outline the extent of necroses in the three examined apple varieties and underscore CE's protective effect against *P. cinnamomi*. It is noteworthy that a varietal susceptibility factor is apparent, as evidenced by the statistically significant differences among the positive controls of 'Golden', 'Starking Delicious', and 'Reinette'. In contrast, the turmeric extract demonstrated notable efficacy, achieving the complete inhibition of mycelial growth in all three instances.

**Table 2.** Results of the Kruskal–Wallis test for necrosis lengths in excised stems of three *Malus domestica* tree varieties, followed by multiple pairwise comparisons using the Conover–Iman procedure between the positive control (inoculated, no treatment), negative control (not inoculated), and treated samples with the *Curcuma longa* extracts (CEs) of each variety (see text for procedure description. Mean rank values accompanied by the same letters are not significantly different (*p*-value (one-tailed) < 0.0001, α = 0.05).

| *M. domestica* Variety | Treatment | Mean of Ranks | Groups | | |
|---|---|---|---|---|---|
| Golden | Positive control | 96.333 | | | D |
| | CE | 30.500 | A | | |
| | Negative control | 30.500 | A | | |
| Starking Delicious | Positive control | 84.633 | | C | |
| | CE | 30.500 | A | | |
| | Negative control | 30.500 | A | | |
| Reinette | Positive control | 68.033 | | B | |
| | CE | 30.500 | A | | |
| | Negative control | 30.500 | A | | |

## 4. Discussion

### 4.1. Phytochemical Profile

Given that the hydroethanolic extraction mixture solubilized polar compounds, which were non-volatile and could not be detected by GC-MS without the prior derivatization of the extract, a note of caution regarding the results is deemed necessary. Furthermore, considering the limitations in identifying certain compounds in the extracts due to the subset of known organic compounds available in the GC-MS databases, especially those with Qual values below 80, caution is advised. It is crucial to acknowledge that, while the identification of these compounds can hold some value, it can also be subject to inaccuracy.

Regarding the previous research on the phytochemical constituents of *C. longa* extracts, Pintatum et al. [33] discovered that, using UPLC–HRMS, compounds like β-sitosterol, curcumenol, curcumin, curdione, dehydrocurdione, germacrone, zederone, and zedoarondiol were common in ethanol extracts of *C. longa*, *Curcuma aromatica* Salisb., and *Curcuma aeruginosa* Roxb. Alvindia et al. [34] identified twenty-three chemical components in *C. longa* crude extract through GC–MS analysis, with *ar*-turmerone (50.63%), curlone (15.42%), α-curcumene (6.48%), and 3-octanol (5.88%) as the four major compounds present. Braga et al. [23], utilizing gas chromatography-flame ionization detection (GC–FID), reported that *C. longa* rhizomes' ethanolic extracts contained *ar*-turmerone, (Z)- and (E)-γ-atlantone as the primary constituents, with *ar*-curcumene, α-zingiberene, β-sesquiphellandrene, and *ar*-turmerol in reduced amounts (<2.4%). Singh et al. [35], via GC–MS, found significant differences in the chemical composition of ethanol oleoresin from the fresh and dry rhizomes of *C. longa*: in the fresh rhizomes, the major components were α-turmerone (53.4%), β-turmerone (18.1%), and aromatic turmerone (6.2%), while in the dry rhizomes, the main chemical species were aromatic turmerone (9.6%), α-santalene (7.8%), and α-turmerone (6.5%). The reported composition aligns more closely with the latter three studies than with the one described by Pintatum et al. [33]. The observed variability in the determined compounds and their relative proportions could be attributed to the fact that compounds from *Curcuma* species could exhibit different responses to various solvents and physical factors during the extraction process [26].

It is worth mentioning that the oxygenated sesquiterpenes and sesquiterpene hydrocarbons identified in CE by GC–MS aligned with those previously reported in essential oils obtained from *C. longa* rhizomes through methods such as steam distillation, hydrodistillation, supercritical fluid extraction, and subcritical water extraction, as summarized in the review by Ibáñez and Blázquez [25]. This is unusual, as hydroethanolic extracts and essential oils typically possess different phytochemical profiles due to the differences in solubility and processing temperature associated with the extraction methods employed

(hydroethanolic extracts, obtained by ethanol soaking of plant material, can extract a broader range of compounds, including polar and nonpolar compounds, and tend to preserve more heat-sensitive compounds, while essential oils consist of volatile and lipophilic compounds). The observed similarities could tentatively be attributed to the utilization of the rotavapor processing step, affirming the appropriateness of the selected extraction procedure. The capability to extract similar constituents through standard hydroethanolic extraction presents several advantages, including reduced volatility, simplified standardization, and enhanced sustainability.

Regarding the main constituents identified in CE by GC–MS, (+)-$\beta$-turmerone is a sesquiterpenoid present in *C. longa*, *Curcuma xanthorrhiza* Roxb. [36], and *Chrysoma pauciflosculosa* (Michx.) Greene [37]. $\alpha$-Turmerone is an enone found in *C. longa* and *C. xanthorrhiza* [37]. (+)-(S)-*ar*-turmerone is another bisabolane sesquiterpenoid that is 2-methylhept-2-en-4-one substituted by a 4-methylphenyl group at position 6 and has also been isolated from *Peltophorum dasyrachis* (Miq.) Kurz [38]. Additionally, $\alpha$-atlantone is a bisabolane-type sesquiterpenoid found in *Zingiber officinale* Roscoe [39]. As for the minor constituents, $\gamma$-curcumene is a bisabolone sesquiterpene also discovered in *Baccharis pedunculata* (Mill.) Cabrera, and *Santolina corsica* Jord. and Fourr. [40]. Zingiberene is a monocyclic sesquiterpene that has been detected in *Chaerophyllum azoricum* Trel., *Helichrysum odoratissimum* (L.) Sw., and in the dried rhizomes of Indonesian ginger (*Z. officinale*), from which it derives its name [41]. Phytochemicals structurally distant from the aforementioned ones are isoelemicin and gibberellin A3. Isoelemicin is a phenylpropanoid found in *Solidago odora* Aiton, *Virola surinamensis* (Rol.) Warb., and *Acorus calamus* L. [42]. Lastly, gibberellin A3 (named after the fungus *Gibberella fujikuroi* (Sawada) Wollenw., 1931) is a pentacyclic diterpenoid that functions as a hormone in the regulation processes of plant development and growth, which include seed development and germination, stem and root growth, cell division, and flowering time [43].

Concerning the structural implications, the benzylic stereogenic center in bisabolanes enables the asymmetric hydrovinylation conversion of 4-methylstyrene into curcumene, *ar*-turmerone, and other identified phytoconstituents [44], all exhibiting subtle stereochemical differences. These closely structured substances in the mixture are expected to have similar biological activities [45]. Therefore, the choice to conduct the activity study not on the individual constituents, but on the crude extract, of which they are the primary elements, is endorsed. This approach can present benefits, such as reduced expenses, increased long-term effectiveness (multi-compound products hinder resistance development, as pathogens struggle to adjust to multiple simultaneous actions), and broader applicability against an array of pests and diseases (compounds frequently work synergistically, offering activity against diverse pathogens, while a single refined product can solely address certain pests, exposing the crop to additional risks).

### 4.2. On the Antimicrobial Activity
#### 4.2.1. Broad Spectrum of the Antifungal and Antioomycete Activities of CE

The antifungal activities of *C. longa* extracts and phytochemicals has been previously documented and recently reviewed [26], specifically against *B. cinerea* [46,47] and *C. acutatum* [48]. However, to our knowledge, there is no reported activity against *D. corticola*, *P. cactorum*, and *P. cinnamomi* in the existing literature. While a hexane extract was found to be active against *Phytophthora infestans* (Mont.) de Bary [46], no information was found regarding the activity against *Diplodia* spp. Therefore, we present, to the best of our knowledge, the first report on the activity of *C. longa* extract against highly devastating phytopathogenic agents, such as *D. corticola*, *P. cactorum*, and *P. cinnamomi*.

In this study, CE exhibited notably different efficacies against the tested pathogens. The pathogen-specific activity of the CE could arise from intricate interactions between the bioactive compounds in the extract and the specific genetic and physiological characteristics of different pathogens. This observation aligns with the findings of Adamczak et al. [49], who, upon testing curcumin's efficacy against over 100 strains of pathogens belonging to

19 species, concluded that curcumin could be considered a promising antibacterial agent, albeit with highly selective activity contingent on the microbial species and strain.

Notably, our results for *F. culmorum* differ from the findings elsewhere [50]. Although CE caused a marked growth inhibition in the initial days of the experiment (approximately 9 days of incubation), the continuous decrease in inhibition over time resulted in only slight differences with the control plate. Presumably, the variations in plant variety, cultivation conditions, and harvest time can account for these differences. Nevertheless, the impact of the particular *F. culmorum* isolate cannot be discounted, as evidenced by the significant variations in MIC values observed by Adamczak et al. [49] among different isolates of yeast-like fungi. Interestingly, the toxic activities of curdione and curcumenol, compounds typically found in *C. longa* extracts, against *Fusarium graminearum* Schwabe involves membrane disruption by inhibiting ergosterol synthesis, respiration, succinate dehydrogenase (SDH), and NADH oxidase [50]. The absence of these compounds in CE suggests that other components can contribute to the activity, and additional toxicity mechanisms and targets are likely to occur with CE.

Considering their abundance in the CE and supported by the previously reported findings, turmerone and its derivatives emerge as the principal constituents contributing to the reported antifungal and anti-oomycete activities. This hypothesis is reinforced by the study conducted by Ferreira et al. [20], demonstrating that an essential oil from *C. longa*, predominantly composed of *ar*-turmerone (33.2%), *α*-turmerone (23.5%), and *β*-turmerone (22.7%), exhibited antifungal activity against *Aspergillus flavus* Link, causing damage to hyphae membranes and conidiophores. Similarly, in a study on *Fusarium verticillioides* (Sacc.) Nirenberg, employing a *C. longa* essential oil containing *α*-turmerone (42.6%), *β*-turmerone (16.0%), and *ar*-turmerone (12.9%), conducted by Avanço et al. [51], these compounds not only reduced the thickness and length of microconidia, but also significantly decreased ergosterol production. Furthermore, Naveen Kumar et al. [52] investigated a *C. longa* essential oil (composed of *ar*-turmerone (53.10%), *β*-turmerone (6.42%), and *α*-turmerone (6.15%)) against *F. graminearum*. Their findings indicated that these compounds permeated the cell through the cell wall and cytoplasmic membrane, disrupting cell membrane integrity. This disruption led to impaired membrane fluidity, cytoplasmic content leakage, and disturbances in osmotic and enzymatic regulations, crucial for ATP synthesis, cell growth, and fungal proliferation.

### 4.2.2. Comparison of Treatment Effectiveness in Excised Stems

Regarding the efficacy of CE as a protective treatment against *P. cinnamomi*, Table 3 presents a comparison with alternative treatments targeting *Phytophthora* spp. Its effectiveness was found to be comparable to that of a conjugated complex of chitosan oligomers (COSs) with an aqueous ammonia extract of *Quercus suber* L. bark [53], and superior to that of the aqueous ammonia extract of *Sambucus nigra* L. flowers [54]. However, it is crucial to note that those treatments were evaluated against *Phytophthora cactorum* and *Phytophthora megasperma*, respectively. Hence, caution is advised when drawing comparisons. In terms of the comparisons specifically involving *P. cinnamomi*, CE exhibited lower effectiveness compared to the aqueous ammonia extract of *Quercus ilex* subsp. *Ballota* (Desf.) Samp. Bark [55] and the conjugate complex of COS with *Ganoderma lucidum* (Curtis.) P. Karst aqueous ammonia extract [31].

Based on the results, CE demonstrates notable toxicity against the oomycete *P. cinnamomi*. This organism is considered one of the ten most destructive and devasting oomycetes known [5], affecting close to 5000 species of plants, including many of importance in agriculture, forestry, and horticulture [11]. Disease control is difficult in agricultural and forestry situations, and even more challenging in natural ecosystems as a result of the scale of the problem and the limited range of effective chemical inhibitors. *Castanea* sp., *Eucalyptus* sp., *Banksia* sp., *Quercus* sp., and avocado plants are reported as the main ones affected by this pathogen [56–58].

**Table 3.** Protective treatments against *Phytophthora* spp. based on natural products.

| Source of Excised Stems | Pathogen | Natural Product | Effectiveness | Ref. |
|---|---|---|---|---|
| *Malus domestica* | | *Curcuma longa* | Full protection at 3000 µg·mL$^{-1}$ | This work |
| *Quercus ilex* | *Phytophthora cinnamomi* | COS—*Ganoderma lucidum* | Full protection at 782 µg·mL$^{-1}$ | [31] |
| *Prunus amygdalus* x *P. persica* | | *Q. ilex* subsp. *ballota* | Full protection at 782 µg·mL$^{-1}$ | [55] |
| | *Phytophthora cactorum* | COS—*Q. suber* | Full protection at 3750 µg·mL$^{-1}$ | [53] |
| | *Phytophthora megasperma* | *Sambucus nigra* | Full protection at 1875 µg·mL$^{-1}$ | [54] |

*4.3. Plant Toxicity*

Lettuce seeds were selected for the evaluation of CE toxicity due to their known susceptibility to toxic compounds, rendering them reliable indicators of toxicity. Additionally, lettuce is recommended by the OECD guidelines for toxicity assessments. While the absence of an effect on seed germination of *Oryza sativa* L. has been reported for *C. longa* essential oil only [59], phytotoxicity in the full development of plants has been observed for an ethanolic extract, particularly in *Lemma minor* L., affecting the development of fronds [60].

The differential response between CE and the literature report can be attributed to the susceptibility of plant species. Nevertheless, considering that seed germination and early plant development represent stages when plants are particularly vulnerable to toxic agents, we can conclude that CE appears to be relatively safe for plants. Phytotoxicity is predominantly associated with the *C. longa* essential oil rather than the hydroethanolic extract.

The study's observations regarding CE, demonstrating no significant impact on early lettuce growth, except for its positive influence on root development, imply potential agricultural benefits. The enhancement of root development can contribute to improved resilience against environmental stressors. CE emerges as a prospective alternative to synthetic pesticides, safeguarding plants from pests or diseases without compromising their overall growth. This reduced dependence on chemical pesticides aligns with the global efforts to promote eco-friendly farming practices, offering environmental advantages, such as decreased soil and water pollution. However, further research, including comprehensive field trials across various crops and conditions, is imperative to validate these preliminary findings and evaluate the practicality of CE in real-world agricultural contexts. Additionally, an exploration of the mechanisms underlying its effects on root growth is warranted.

**5. Conclusions**

The hydroethanolic extract of *C. longa*, identified by GC–MS to contain (+)-$\beta$-turmerone bisabolene sesquiterpenoid along with five known analogs, exhibited a phytochemical family with a skeletal structure and functional groups consistent with the information elucidated using ATR–FTIR. Concerning its antimicrobial activity, in this study, conducted on a range of phytopathogenic microorganisms (*B. cinerea*, *C. acutatum*, *F. culmorum*, and, for the first time in the literature, *D. corticola*, *P. cactorum*, and *P. cinnamomi*), the extract demonstrated its highest efficacy against *P. cinnamomi*, a destructive oomycete affecting tree crops. Ex situ assays conducted on excised *M. domestica* stems revealed complete protection against *P. cinnamomi* at a concentration of 3000 µg·mL$^{-1}$. Furthermore, tests on *L. sativa* suggested that the extract could have low or absent phytotoxicity. Hence, this study shows that CE presents a promising approach for mitigating microbial threats to crops, particularly in safeguarding tree crops against *P. cinnamomi*. Its environmentally friendly characteristics position it as a valuable asset in the pursuit of sustainable agriculture.

**Supplementary Materials:** The following supporting information can be downloaded at: https://www.mdpi.com/article/10.3390/horticulturae10020124/s1, Figure S1: infrared spectrum of the *C. longa* hydroethanolic extract; Figure S2: chemical structures of phytochemicals identified in the hydroethanolic extract of *Curcuma longa* by GC–MS; Figure S3: *Malus domestica* cv. 'Golden', 'Starking Delicious', and 'Reinette' stem segments artificially inoculated with *P. cinnamomi* after 4 days of incubation, untreated samples vs. samples treated with *C. longa* hydroethanolic extract at a 3000 μg·mL$^{-1}$ dose; Table S1: main bands in the infrared spectrum of the *Curcuma longa* hydroethanolic extract and their assignments; Table S2: MS data for the hydroethanolic *C. longa* extract.

**Author Contributions:** Conceptualization, P.M.-R. and R.O.; methodology, P.M.-R., A.C. (Ana Cunha) and R.O.; validation, P.M.-R. and R.O.; formal analysis, P.M.-R. and R.O.; investigation, A.C. (Adriana Cruz), E.S.-H., A.T., P.M.-R., A.C. (Ana Cunha) and R.O.; resources, P.M.-R. and R.O.; writing—original draft preparation, A.C. (Adriana Cruz), E.S.-H., P.M.-R. and R.O.; writing—review and editing, E.S.-H., P.M.-R. and R.O.; visualization, A.C. (Adriana Cruz), E.S.-H. and A.T.; supervision, P.M.-R. and R.O.; project administration, P.M.-R. and R.O.; funding acquisition, P.M.-R. and R.O. All authors have read and agreed to the published version of the manuscript.

**Funding:** This work was supported by the FCT—Portuguese Foundation for Science and Technology, under projects UIDB/04050/2020 and UIDB/04033/2020, by AgrifoodXXI (NORTE-01-0145-FEDER-000041, and by Junta de Castilla y León under project VA148P23, with FEDER co-funding.

**Data Availability Statement:** The data supporting the findings of this study are available within the article and its Supplementary Materials.

**Acknowledgments:** Thanks to Richard Breia and Hernâni Gerós (Department of Biology, University of Minho, Braga, Portugal), Pedro Talhinhas (School of Agriculture, University of Lisbon, Lisbon, Portugal), Ana Cristina Esteves (Centre for Environmental and Marine Studies, CESAM, University of Aveiro, Aveiro, Portugal), Aldearrubia Regional Diagnostic Center—Junta de Castilla y León (Salamanca, Spain), and the Calabazanos Forest Health Center—Junta de Castilla y León (Palencia, Spain), for providing the isolates used in the study.

**Conflicts of Interest:** The authors declare no conflicts of interest. The funders had no role in the design of the study; in the collection, analyses, or interpretation of data; in the writing of the manuscript; or in the decision to publish the results.

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
