# Peer review of "Antifungal and Antioomycete Activities of a Curcuma longa L. Hydroethanolic Extract Rich in Bisabolene Sesquiterpenoids"

_horticulturae, doi:10.3390/horticulturae10020124_

Round 1

Reviewer 1 Report

Comments and Suggestions for Authors

This original and groundbreaking research, which explores the antimicrobial potential of C. longa extracts, particularly CE, against various phytopathogenic microorganisms, merits publication for its significant contributions to the field of agricultural science and sustainable crop protection. However, to further enhance the quality of the article, here are my constructive suggestions:

Introduction section:

- The introduction provides detailed background information, which is beneficial. However, to maintain reader engagement, consider condensing some parts, particularly the chemical composition of turmeric. Focus more on its relevance to the study's aim.

- Ensure a logical flow of ideas. Start with the broader issue (global agricultural challenges), then narrow down to specific problems (plant diseases, the role of pesticides, environmental concerns), before introducing your research focus (use of natural compounds, specifically turmeric).

- While providing extensive background information, ensure that every point directly relates to your research focus. For instance, explicitly link the detailed composition of turmeric to its potential role in agriculture.

Materials and Methods section:

  • In line 135, the process of sample preparation for both analyses is clear. However, include the rationale for using methanol in GC-MS analysis, especially if it's important for the solubility of the compounds in your extract.
  • In line 165, you've appropriately included an 80% (v/v) ethanol solvent control. Clarify if this control was used to account for any potential effects of the solvent itself on mycelium growth.
  • In line 181, the reference to the procedure outlined by Matheron and Mircetich with modifications is good. Consider summarizing the key steps of this procedure directly in the text, especially the modifications, for the convenience of the reader.

3.2. In Vitro Antimicrobial Activity:

  • Provide interpretation or discussion about why CE has different effects on different pathogens. Are there any specific properties of CE that might explain this variation?
  • Emphasize the novelty of your findings, especially the activity of CE against D. corticola, P. cactorum, and P. cinnamomi. This adds value to your research, highlighting its contribution to expanding current knowledge.
  • The difference in findings with F. culmorum compared to other studies is an interesting point. Discuss potential reasons for these differences in more detail, considering factors like extract composition, concentration, and experimental conditions.
  • The discussion on the toxic activity of specific compounds like curdione and curcumenol against Fusarium graminearum is insightful. It would be beneficial to hypothesize about the compounds in CE that could be responsible for the observed effects, based on their known bioactivities and the GC-MS analysis results.
  • In the final paragraph, you discuss the absence of certain compounds in CE compared to other C. longa extracts. Consider elaborating on what this means for the potential toxicity mechanisms and targets of CE. Are there any hypotheses or suggestions based on your results?

3.3. Germination and Early Growth of Lettuce Seedlings:

  • Discuss the implications of these findings in the context of agricultural practices. For example, is CE a viable option for protecting crops without affecting their growth?
  • Mention any potential environmental benefits, such as reduced reliance on synthetic chemicals.

Conclusion section:

  • End with a strong concluding statement that summarizes the overall significance of your research. Suggested addition: "In summary, CE offers a promising avenue for addressing microbial threats to crops, and its environmentally friendly characteristics make it a valuable asset in the pursuit of sustainable agriculture."

Reviewer 2 Report

Comments and Suggestions for Authors

The manuscript "Antimicrobial Activity of a Curcuma longa L. Hydroethanolic Extract Rich in Bisabolene Sesquiterpenoids" is interesting and the data presented in it may have great application potential. However, it seems to me that the chemical analysis of the extract used is insufficient. The IR analysis was not needed here and does not provide relevant data (Tab. 1 should be pzrenies to the supplementary material). The GC-MS analysis is well done however I would have expected a chromatogram with numbered peaks and a table with mass data in the main body of the manuscript. I also believe that the authors should additionally perform HPLC analysis because non-volatile compounds with high biological activity are present in the ethanol-water extract (doi: 10.1002/ptr.6035 ; https://doi.org/10.3390/ijms20040898 ; 10.3390/ijms23020639). The extract is probably available to the Authors and characterization of its polyphenolic profile should not be a problem. Then the discussion will be much more interesting.

Reviewer 3 Report

Comments and Suggestions for Authors

The manuscript aimed to assess the antifungal activity and plant protection capabilities of a hydroethanolic extract of C. longa as a natural product against crop pathogens.

The references are inherent and recent, but a high number of self-references were found. The manuscript is well-structured and well-written, and it seems to fit the journal scope. However, there are some things to improve.

General comments
The activity of the pure compounds found in the extract should be tested alongside the activity of the pure compounds. Lines 362-368: Without tests on pure compounds this part turns out to be speculation. Lines 177-180 reported a concentration of 3000 µg ml 1 which seems a little high to define a biological activity. In fact, Table 3 reports lower concentrations for the same pathogen treated with other substances.

Specific comments

Line 19: the abbreviation of “hydroethanolic CE” in the abstract should be avoided. 

Comments on the Quality of English Language

 Minor editing of English language required

Reviewer 4 Report

Comments and Suggestions for Authors

Dear authors, I reviewed in detail the paper entitled " Antimicrobial Activity of a Curcuma longa L. Hydroethanolic Extract Rich in Bisabolene Sesquiterpenoids". 

These are my comments and suggestions:

The title should be more precise: “Antifungal and antioomycete activities of a Curcuma longa L. Hydroethanolic Extract Rich in Bisabolene Sesquiterpenoids" is more appropriate.

In section 2.2. write the concentration of stock solutions.

In Figure 1, the graphs should be in color and the microorganism should be written above each graph for better visibility and easier monitoring of the results.

In Figure 2, the graphs also should be in color for better visibility of the results.

Please match the references with the journal instructions, especially the names of the journals (abbreviations).

The manuscript is interesting, very well structured, interesting results and well presented, thoroughly written, English is understandable. I suggest acceptance after accepting all comments. I suggest acceptance after minor revision.

Kind regards

Round 2

Reviewer 2 Report

Comments and Suggestions for Authors

The authors followed my comments and made the necessary corrections. I understand that they may have a problem accessing HPLC, especially in the time required to make corrections.   However, the authors have pointed out the limitations of the study which makes the manuscript fully objective. I believe it can be accepted for publication.